# Mesenchymal Stem Cell Therapies Approved by Regulatory Agencies around the World

**DOI:** 10.3390/ph16091334

**Published:** 2023-09-21

**Authors:** Luis E. Fernández-Garza, Silvia A. Barrera-Barrera, Hugo A. Barrera-Saldaña

**Affiliations:** 1Laboratorio Nacional de Servicios Especializados de Investigación, Desarrollo e Innovación de Farmoquímicos y Biotecnológicos (LANSEIDI) del CONACyT, Sede Innbiogem SC, Monterrey 64630, Mexico; luisfdz12@gmail.com (L.E.F.-G.); silviaabb92@gmail.com (S.A.B.-B.); 2Departamento de Medicina Interna, Hospital General de Zona con Medicina Familiar No. 2 del Instituto Nacional del Seguro Social, Monterrey 64010, Mexico; 3Facultades de Medicina y Ciencias Biológicas, Universidad Autónoma de Nuevo León, San Nicolás de los Garza 66455, Mexico; 4Columbia Investigación Científica, Panzacola 62, Colonia Villa Coyoacán, Alcaldía Coyoacán, Ciudad de Mexico 04010, Mexico

**Keywords:** mesenchymal stem cells, approved therapies, cell therapy medical products, regulatory agencies

## Abstract

Cellular therapy has used mesenchymal stem cells (MSCs), which in cell culture are multipotent progenitors capable of producing a variety of cells limited to the mesoderm layer. There are two types of MSC sources: (1) adult MSCs, which are obtained from bone marrow, adipose tissue, peripheral blood, and dental pulp; and (2) neonatal-tissue-derived MSCs, obtained from extra-embryonic tissues such as the placenta, amnion, and umbilical cord. Until April 2023, 1120 registered clinical trials had been using MSC therapies worldwide, but there are only 12 MSC therapies that have been approved by regulatory agencies for commercialization. Nine of the twelve MSC-approved products are from Asia, with Republic of Korea being the country with the most approved therapies. In the future, MSCs will play an important role in the treatment of many diseases. However, there are many issues to deal with before their application and usage in the medical field. Some strategies have been proposed to face these problems with the hope of reaching the objective of applying these MSC therapies at optimal therapeutic levels.

## 1. Introduction

In Europe, emerging new gene and cell therapies have led to the creation of laws that regulate these products commonly known as advanced therapy medicinal products (ATMPs). ATMPs are divided into four types:Cell therapy medicinal products (CTMPs). These contain manipulated cells or tissues that are intended to be used for different essential functions in the recipient, compared to the one that they had in the donor.Tissue-engineered products (TEPs). Engineered cells or tissues that can regenerate, repair, or replace human tissue.Gene therapy medicinal products (GTMPs). Related to diagnostic, therapeutic, or prophylactic effects that use specific recombinant nucleic acids.Combined ATMPs (cATMPs). These integrate two or more of the products listed above [1].

Each regulatory agency around the world has its own product classification with small differences between them. Some of the regulatory agencies around the world are the European Medicines Agency (EMA), the Food Drug Administration (FDA) in the United States of America (USA), the Ministry of Food and Drug Safety (MFDS) in Republic of Korea, the Drug Controller General of India (DCGI), the Therapeutic Goods Administration (TGA) in Australia, and the Pharmaceuticals and Medical Devices Agency (PMDA) in Japan.

The previous regulatory agencies were mentioned because they have until now been the only ones that could approve CTMP therapies that use their originally baptized mesenchymal stem cell name (commonly referred to as MSCs); these agencies have attempted to change their name to medicinal signaling cells, and more recently to mesenchymal progenitors (Arnold I. Caplan, personal communication; see Acknowledgement statement). For convenience, the term MSCs will be used from here on [2,3,4].

Going back to the basics, stem cells, in general, are classified based on their differentiation capacity including totipotent, pluripotent, multipotent, and unipotent qualities. Totipotent stem cells have the potential to self-renew by dividing themselves and to differentiate into any type of cell found in an organism, which is the case of the zygote egg cell. Pluripotent stem cells have the potential to self-renew by dividing and to give rise to all the cells in the mesoderm, endoderm, and ectoderm layers, but not extra-embryonic tissues such as the placenta. Then, the previous cells give rise to different types of specialized cells in the body becoming multipotent. Multipotent stem cells can produce a variety of cells limited to the germinal layer of origin, with this being the case for MSCs, that have the potential to differentiate into muscle, bone, and cartilage tissues, all derived from the mesoderm. Lastly, unipotent stem cells create only one lineage of cells, for example, those of muscle or skin can only give rise to muscle or skin cells, respectively [5]. Another way to classify them is based on the developmental stage in which they can be extracted, such as embryonic, fetal, infant, or adult stem cells [5]. Embryonic stem cells are derived from the inner cells of a blastocyst; these are highly undifferentiated and highly proliferative and can comprise totipotent and pluripotent stem cells. Fetal stem cells, which include MSCs as the name implies, can be isolated from fetal tissues such as blood, bone marrow (BM), liver, and kidney. Infant MSCs include those extracted from the umbilical cord blood, amnion, or placenta. Lastly, adult MSCs include those isolated from adipose tissue, BM, peripheral blood, and tooth pulp [6] (Figure 1). 

We have carried out a systematic review to analyze the different applications that MSCs have had in various registered clinical trials around the world [7]. Now, in this narrative review, we address the information about current MSC therapies approved by regulatory agencies, convey the problems that these therapies usually face, and propose solutions to overcome these problems.

## 2. Mesenchymal Stem Cells

As mentioned above, the terms medicinal signaling cells and mesenchymal progenitors have started to be used by experts in the field to replace previous names to refer to MSCs [2,3,4]. Nevertheless, we are going to use the latter to avoid confusion. MSCs come from cells derived from the embryonic mesoderm and they are capable of self-renewal, and under appropriate stimulation, of differentiating into various tissues in vitro, including bone, cartilage, muscle, BM stroma, tendon/ligament, fat, and dermis [8,9,10]. On the other hand, MSCs in vivo (or in physio, referring to what occurs inside the body) originate as perivascular cells and do not differentiate into other cell types to repair damaged tissue, but rather, they detach from the blood vessels of the damaged tissues to generate a regenerative microenvironment through their immunomodulatory secretions and trophic bioactive factors, inducing other cells (stem cells from the damaged tissue) to regenerate the tissue [11,12,13]. It was precisely for this property that they were proposed to be renamed medicinal signaling cells [4]. 

Adult bone marrow MSCs (BM-MSCs) are the most widely used in clinical trials; however, MSCs from birth-associated tissues may have additional biological properties, such as high proliferative capacity, longer life span, and greater differentiation potential [14]. Depending on their origin, these cells exhibit different immunophenotypes that explain some of the variations in their responses. Adult MSCs positively express CD44, CD90, CD105 (SH2), and CD166, and negatively express CD14, CD34, and CD45 [15]. Birth-associated MSCs express more primitive surface markers. Moreover, MSCs from the amnion membrane and umbilical cord cannot conduct in vitro adipogenic differentiation, while those from the placenta and umbilical cord express genes for hematopoietic growth factors that give them the ability to develop into hematopoietic progenitor cells [7]. 

MSCs can be administered from autologous or allogeneic sources [7]. BM-MSCs are obtained from the BM extracted from the iliac crest, femur, or tibia. The isolation of MSCs from the BM can be performed via density gradient centrifugation or direct plating. The density gradient centrifugation method involves the use of gradient density solutions with low viscosity and low osmotic pressure which allows for the separation of the different types of cells depending on their density. Additionally, in the direct plating method, cells are seeded into a low-density culture dish, approximately 10³ to 10⁶ cells/cm², separated by adhesin, and finally subjected to expansion. The MSCs expanded in culture do not lose their differentiation potential, although their lifespan is limited to 15–50 population doublings [8,16]. MSCs from adipose tissue (AT-MSCs) are obtained from liposuction or lipectomy samples (visceral or subcutaneous adipose tissue from the abdomen, brachium, femoral, or gluteal areas). AT-MSC isolation typically involves the enzymatic digestion of adipose tissue samples by collagenases, followed by the removal of red blood cells (RBCs) using a specific RBC lysis procedure and filtration of the cells [8]. MSCs from neonatal tissues can be obtained from the entire umbilical cord, including the connective tissue, Wharton’s jelly, and the blood vessels. The cell biology methods used to isolate MSCs from the umbilical cord vary depending on which compartment of the umbilical cord is selected as the extraction point. Typically, these methods may include enzymatic digestion of umbilical cord tissue samples, RBC-specific lysis, cell filtration, and/or density gradient separation. The isolation/expansion procedure is similar for adult and neonatal MSCs [8].

It has been speculated that the heterogeneity of MSCs reflects the diversity of the environments from which they can be isolated; all of this is due to the propagation difficulties in vivo, which have been established in in vitro studies. Despite comparing different donors, it has been observed that the functionality of MSCs decreases as time goes by. In turn, MSCs isolated from the same donor and the same source at different time points also show significant variation in growth rates. A decrease in the function of MSCs has been confirmed over time, concerning proliferation, colony formation, telomere length, and differentiation capacity. Regarding the same, the secretion of bioactive molecules with the paracrine effect also decreases as the number of cell divisions increases. All of this leads to a loss of the therapeutic potential that the following clones of MSCs will have. This has been confirmed in a study that showed that after the twelfth division, clonal complexity decreased considerably, resulting in a weakly dominant clonal selection. All this in turn is influenced by the microenvironmental conditions that the MSCs face in vivo. Factors such as the composition of the matrix, its geometry, mechanical properties, oxygen concentration, nutrients, and interactions with other cells included in the microenvironment can lead to mutations or cell defects that favor the heterogeneity of MSCs [17].

Another thing to take into account among its characteristics is the MSC transcriptome, which includes a large repository of gene-expression-based data, and the expression changes that happen after culture expansion, hypoxia preconditioning, stimulus-directed differentiation, trans-differentiation, exposure to biologics, and coculture with other cell types [18]. Despite the fact that there is still much to be clarified in this matter, the transcriptome of human and mouse BM-MSCs was cataloged via serial analyses of gene expression, and the nature of the cataloged transcripts appeared to reflect their stem/progenitor properties and paracrine activities related to skeletogenic, angiogenic, anti-inflammatory, and immunomodulatory properties. MSCs isolated and cultured from distinctive tissues/organs are more closely related to each other than other mesodermal heredities; also, their phenotypic signature is comparable to that of perivascular cells, giving a physiological premise for the far-reaching anatomical dissemination of MSCs in vivo. The degrees to which MSC gene expression is impacted by their culture conditions and can be controlled remain critical for their clinical utility [18]. RNA-seq and ChIP-Seq studies have revealed how MSCs react at the cellular level, inducing differentiation boosts, and how the epigenome of MSC-derived osteoblasts, but not adipocytes, closely resembles that of naive cultures of MSCs, respectively. Regardless of the therapeutic MSC products, a release assay related to the clinical proposed function of the MSCs in vivo is requested by the FDA in Phase 1 and also in Phase 3; these RNA-Seq and ChIP database kits ought to be used more to match appropriate donor MSCs, manufacturing protocols, and patients to improve response rates [18]. 

## 3. Clinical Applications of MSCs

We performed a systematic review up until June 2020; the findings comprised 903 registered clinical trials at clinicaltrials.gov using MSCs as treatments, which had complete information about their objective and current phases. The medical specialties with the largest number of registered clinical trials were orthopedics, pneumology, neurology, cardiology, and immunology (from highest to lowest). Concerning the phases in which these clinical trials were found, 278 were in Phase 1, 551 were in Phase 2, 68 were in Phase 3, and the remaining 6 were in Phase 4. Despite many registered studies, only 18 of them had published results [7]. 

Regarding these published results, most of the diseases for which MSCs were being used did not have a cure. For example, in the field of orthopedics, cartilage lesions and osteoarthritis stand out. Currently, the treatments for these illnesses are hard drugs or surgical measures to reduce discomfort and pain. 

Two treatment strategies using MSCs were published reporting clinical improvements in pain, stiffness, and functionality in patients with knee osteoarthritis. One of these uses an intra-articular injection of autologous or allogeneic BM-MSCs and another replaces the defect in the cartilage with a fibrin glue combined with chondronds and allogeneic BM-MSCs [7]. In cardiology, promising results in clinical and pathophysiological outcomes were found in diseases such as dilated cardiomyopathy and ischemic or non-ischemic heart failure. In this case, MSCs were delivered either locally (intramyocardial, intracoronary, and trans-endocardial) or intravenously, with an autologous or allogeneic source, and based on BM-MSCs, they showed improvements in cardiac function, functional capacity, and quality of life [7]. 

In pneumology, two published trials, with one about idiopathic pulmonary fibrosis, with allogeneic MSCs from the placenta, related or unrelated to a human leukocyte antigen (HLA)-identical or HLA-mismatched donor, were used. These patients were able to walk longer distances and had an improvement in lung function. The other pathology was acute respiratory distress syndrome, for which allogeneic BM-MSCs were used. Unfortunately, this trial did not show any benefits [7]. Concerning neurology/neurosurgery, spinal cord injury was treated through intrathecal and subarachnoid autologous BM-MSC injections. Both trials resulted in improvements in sensitivity, pain, neurophysiological parameters, functional independence, and urodynamic studies [7]. In the rest of the areas, AT-MSC-enriched fat grafts were injected in patients that required breast reconstruction, showing promising results with volume persistence [7]. On the other hand, MSCs were used to treat sickle cell disease and advanced glaucoma, without good results in the trials [7].

In addition to the direct administration of MSCs, there has been an increasing number of applications involving the administration of their derivatives, whether they are pleiotropic factors or extracellular vesicles, which have clinical trials already underway. The MSC-derived pleiotropic factor (MSC-PF), also called secretome, is a rich and complex set of molecules secreted from living cells or shed from the cell surface [19]. In the case of MSC-PF, they include factors such as IGF-1, vascular endothelial growth factor (VEGF), transforming growth factor-beta 1 (TGF-B1), and hepatocyte growth factor (HGF). They have been used for the treatment of different types of wounds, including endonasal wounds, post-surgical wounds, wounds with poor healing, and burns [20,21,22,23]. MSC extracellular vesicles (MSC-EVs), or more specifically, MSC exosomes, are a type of vesicle that cells secrete into the extracellular space, and two important functions have been attributed to these: the capacity for intracellular communication and the transport of RNA [24]. Different forms of administration are being investigated, including classic forms through injections, as well as newer forms that include nebulization, aerosols, or even eye drops. On the other hand, they are being investigated for a wide variety of applications; among which, acute type A aortic dissection, decompensated liver cirrhosis, acute ischemic stroke, degenerative meniscus injury, Alzheimer’s disease, dry eye diseases, and post-refractive surgery stand out, and they are associated with blepharospasm, retinitis pigmentosa, refractory ulcerative colitis, refractory Crohn’s disease, perianal fistulas, epidermolysis bullosa, burn wounds, and even aesthetic treatment for skin rejuvenation. Regarding COVID-19, MSCs have been applied from mild to moderate to ARDS-associated COVID-19, and even the most recently described chronic post-COVID-19 syndrome. It should be noted that this treatment has not only been tried in adults but even in preterm neonates at high risk of bronchopulmonary dysplasia (BPD) [25,26].

## 4. Agency-Approved MSC Therapies

Most treatments usually receive approval from regulatory agencies after Phase 3, making them commercially available to continue with Phase 4. By April 2023, three years after our systematic review, the number of registered clinical trials had increased from 903 to 1120 clinical trials. However, the number of approved products by regulatory agencies around the world remains the same, with only twelve (Table 1). In 2010, the first product that contained MSCs was approved by the MFDS or the Korean Food and Drug Administration (KFDA) in Republic of Korea. The product is named Queencell and it is manufactured by Anterogen. Queencell is composed of stromal vascular fraction (SVF), which is a by-product of the harvesting of excess fatty tissue. SVF comprises a heterogeneous mixture of AT-MSCs and other cell types, such as preadipocytes, endothelial progenitor cells, mast cells, and fibroblasts; it was approved for the treatment of subcutaneous tissue defects [27].

Nine of the twelve approved MSCs products are from Asia. The country in this region with the most approved products is Republic of Korea, with five of them. These include the previously referred to Queencell, in addition to Cellgram-AMI, Cupistem, Cartistem, and Neuronata-R. Cellgram-AMI is a method to improve the left ventricular ejection fraction in patients with acute myocardial infarction (AMI) reperfused by coronary angioplasty within 72 h after chest pain [28]. Cupistem showed that 82% of patients with complex Crohn’s fistula achieved complete healing in eight weeks after treatment and 81% of them sustained a positive response up to Week 96 [27]. Cartistem is used for the treatment of knee cartilage defects associated with osteoarthritis grade IV (the most severe stage of the classification with no visible cartilage remaining), recommended by the International Cartilage Repair Society (ICRS) [29]. Finally, NeuroNata-R has a neuroprotective effect that relieves the progression of amyotrophic lateral sclerosis (ALS) through the extension of motor nerve cell survival due to its anti-inflammatory and immunomodulatory effects [30].

With regard to the other four remaining products, two are from Japan, one is from India, and one more is from Iran. The products in Japan are Temcell HS and Stemiral. Temcell HS is an approved therapy for acute and refractory graft versus host disease (GVHD) in children and adult patients undergoing BM transplants for hematologic malignancies [31]. Stemirac was the first stem cell treatment approved for spinal cord injuries [32]. In India, Stempeucel was approved for the treatment of critical limb ischemia (CLI) due to Buerger’s disease and peripheral arterial disease [33]. Lastly, in Iran, Mesestrocell is used as an osteoarthritis treatment [34].

In Europe, the EMA has only approved two MSC therapies so far. Holoclar is used in the eye to replace damaged cells in the cornea epithelium in adults to treat moderate to severe limbal stem cell deficiency caused by burns [35]. Alofisel was approved for the treatment of complex anal fistulas in adult patients with Crohn’s disease [36]. Finally, in the USA, the country with the most MSC clinical trials conducted, the only approved MSC therapy so far is Remestemcel-L, which is an allogeneic cell product used for the treatment of steroid-refractory acute GVHD in pediatric patients [37].

**Table 1 pharmaceuticals-16-01334-t001:** Current MSC-approved treatments.

Name/Year	Company	Origin	Use	Description	Dosage	AT/AL	Charge
Queen-cell/2010 [1,27]	Anterogen (Republic of Korea)	Republic of Korea MFDS	Connective tissue disorders	A heterogeneous mixture of AT-MSCs and other cell types such as pre-adipocytes, endothelial progenitor cells, pericytes, mast cells, and fibroblasts	1 × 10⁶ cells/mL	AT	NA
Cellgram AMI/2011 [1,28]	Pharmicell (Republic of Korea)	Republic of Korea MFDS	AMI	Human BM-MSCs	a. Under 60 kg = 10 mL/5.0 × 10⁷ BM-MSCs b. 61–80 kg = 14 mL/7.0 × 10⁷ BM-MSCs c. >80 kg = 18 mL/9 × 10⁷ BM-MSCs	AT	USD 15,000 per shot
Cupistem/2012 [1,27,38]	Anterogen (Republic of Korea)	Republic of Korea MFDS	Crohn’s fistula	Human-adipose-tissue-derived MSCs	Fistula diameter: (a) ≤1 cm (3 × 10⁷ MSCs in 1 mL) (b) 1 < X< 2 cm (6 × 10⁷ MSCs in 2 mL)	AT	USD 5000 per treatment
Cartistem/2012 [1,29]	Medipost (Republic of Korea)	Republic of Korea MFDS	Knee OA (ICRS grade IV)	Human-umbilical-cord-blood-derived MSCs	7.5 × 10⁶ cells/vial (depending on the size of the lesion)	AL	USD 21,000 per treatment
NeuroNataR/2014 [1,30,39]	Corestem (Republic of Korea)	Republic of Korea MFDS	ALS	Human BM-MSCs	1 × 10⁶ BM-MSCs/kg Twice every 4 weeks	AT	USD 55,000 annually
Holoclar/2015 [35,40]	Chiesi Farmaceutici	EMA	Limbal stem cell deficiency due to ocular burns	Limbal stem cells	79,000–316,000 cells/cm²	AT	USD 80,000 per eye
Reme-stemcel-L/2015 [1,5,8]	Mesoblast, Ltd. (Australia)	US FDA	Acute and refractory GvHD for pediatric patients	Human BM-MSCs	IV administration: Low (2 million cells/kg) High (8 million cells/kg)	AL	USD 200,000 per treatment
Temcell HS/2015 [1,31,41]	JCR Pharmaceuticals (Japan)	Japan PMDA	Acute and refractory GvHD	Human BM-MSCs	IV infusion of 2 million cells/kg (each bag contains 72 million cells in 18 mL of saline); 4 mL per minute twice weekly in intervals of 3 days or more for 4 weeks	AL	USD 7600 per bag
Stem-peucel/2016 [1,33]	Stem-peutics Research Bangalore (India)	India DCGI	CLI	Human BM-MSCs	Intramuscular injection of 1 or 2 million cells/kg body	AL	USD 2200 per treatment
Alofisel/2018 [1,36,42]	TiGenix (US) and Takeda (UK)	EMA	Complex perianal fistulas in CD	Human-adipose-tissue-derived MSCs	Vial: 30 million MSCs/6 mL Treatment: 4 vials	AL	USD 47,485 per treatment
Mesestro-Cell/2018 [1,34]	Cell Tech Pharmed (Iran)	Iran FDA	OA	BM-MSCs	A minimum intra-articular injection of 2 × 10⁷ cells/knee; in total, 4 × 10⁷ cells for both knees	AT	NA
Stemirac/2018 [32,43]	Nipro Corp	Japan PMDA	Spinal cord injury	BM-MSCs	50 to 200 million cells	AT	USD 135,000

AL: allogeneic, ALS: amyotrophic lateral sclerosis, AMI: acute myocardial infarction, AT-MSCs: MSCs from adipose tissue, AT: autologous, BM: bone marrow, CD: Crohn’s disease, CLI: critical limb ischemia, DCGI: Drug Controller General of India, EMA: European Medicine Agency, FDA: Food and Drug Administration, GvHD: graft versus host disease, ICRS: International Cartilage Repair Society, IV: intravenous, MFDS: Ministry of Food and Drug Safety, MSCs: mesenchymal stem cells, NA: not available, OA: osteoarthritis, PMDA: Pharmaceuticals and Medical Devices Agency, UK: United Kingdom, US: United States.

## 5. Current Issues and Future Directions

MSCs will play an important role in the treatment of many diseases and conditions that currently do not have an effective treatment or cure. Although it is theoretically possible to transfer MSCs from the bench to the bedside, significant failures in quality control and many inconsistencies (regarding immunocompatibility, stability, heterogeneity, differentiation, and migratory capacity) have been reported in several clinical trials [44]. Even though the product has been approved, the manufacturing company continues to face the possibility of having it withdrawn from the market. This could be due to problems with the safety of the product, or not applying or paying for a renewal of the marketing authorization after the agreed deadline. This problem is more common in products used to treat diseases where the incidence is very low and the price of the treatment is very high [45]. Also, another considerable economic problem that faces these therapies in their applicability is that most insurance companies consider MSC therapies to be experimental, and they do not cover their costs.

Apart from the problems mentioned above, the most relevant obstacle that these products are presented with is being accepted by regulatory agencies, especially by the FDA, which is the agency with the fewest accepted products of this type. Part of the legacy-driven requirements for FDA approval have been set up in the last 50 years as a result of the interface of pharmaceutical companies, and this criterion is quite suitable for small molecule drugs but is not suitable for cell-based therapies at this stage of our technological competency. An approach recently proposed by world-renowned MSC expert, Dr. Arnold I. Caplan, is to configure the FDA approval criteria based on the mechanism of action of the therapies being tested (Arnold I. Caplan, personal communication; see Acknowledgement statement). The mechanism of action of cell-based therapies is confusing, because, once an MSC enters the bloodstream of the patient being treated, it is difficult to determine where it docks, what it secretes in the bloodstream or at the docking site, and what cascade of events its presence in the bloodstream or at a particular site will trigger. Even though there are studies that label and track cells through imaging studies such as magnetic resonance imaging, nuclear medicine imaging, and optical imaging, their results have only been validated in preclinical studies [46]. The use of MSCs in clinical trials is based on the capabilities that have been shown in in vitro studies. Among these capabilities are some of the ones that have already been discussed, but many others are worth mentioning as they are released from perivascular locations, surveyed, and sensed, such as that they respond to their microenvironments, modulate the immune system, secrete pro- or anti-inflammatory molecules based on the environment, offer immuno-evasive actions, manage pain by secreting molecules that occupy opioid receptors, secrete mitogenic molecules to tissue-intrinsic committed progenitors, secrete angiogenic molecules, secrete proteins that are antibacterial and antiviral, inhibit scar formation, suppress microglial activation and modulate neuroinflammation, modulate T-cell proliferation and suppress systemic inflammation, and are eaten by Ly6Clow monocytes which then change T-cells to regulatory T-cells which can account for long-term therapy [47,48,49,50]. The need for the FDA to require the mechanism of action of these cells for the approval of any clinical trial is out of context with the reality of their use and of their proven exhibited safety within a medical context.

Currently, there are animal studies that include several strategies that have been proposed to solve part of these problems. Among them are genetic modifications or priming strategies to modify the inherent characteristics of MSCs, biomaterial strategies to modify the outside circumstances, and the usage of MSCs’ secreted proteins and peptides (secretome). The genetic modification includes viral DNA transduction, mRNA/DNA transfection, or more recently, CRISPR-Cas9 technology, with higher effectiveness and specificity [51]. The priming of MSCs could be conducted with pro-inflammatory mediators (IFN-y, TNF-a, IL-1a, IL-1b, IL-17a, and LPS), with hypoxia (1 to 5% of O_2_), or with biomaterials (3D cell culture in collagen-hydrogel scaffold or chitosan scaffold, high-glucose-concentration culture medium, spheroid formation, and matriline-3-primed spheroids) [44]. In the case of biomaterials, they are used to improve the delivery of MSCs, showing benefits in their adherence and survival [44]. Lastly, the collection of proteins and peptides in the secretome of MSCs includes growth factors, cytokines, chemokines, and even extracellular vesicles (EVs). This secretome has been shown to improve the safety profile and the efficacy in the treatment of neurological diseases with MSCs, due to the greater ability that it provides to the cells for crossing the blood–brain barrier and blood–retinal barrier [52].

While many clinical trials are underway and the various pathologies to test its efficacy as a treatment are being explored, we come to another crossroad. Many people, including medical staff, want to take advantage of the desperate situation in which patients find themselves and set up businesses selling products that have not been approved, including some that are not even in clinical studies. A recent report, which only includes the USA, reports that there are 1480 businesses and 2754 clinics selling MSC treatments. This is four times higher than that of five years ago [45]. Another very similar problem is so-called medical tourism, which consists of people usually going abroad to underdeveloped countries in search of treatments that are not approved or are even banned in their country [53]. Most of these clinics offer treatments for multiple diseases for which MSCs have not been approved yet, such as autism spectrum disorders, alopecia, anti-aging therapies, chronic pain, erectile dysfunction, and even coronavirus disease 2019 (COVID-19) [54]. In addition to exploiting the suffering, hope, fear, and economy of patients, these treatments represent a potential risk to their health. There are several potential safety concerns reported, including site reactions, the ability of cells to move from the site of application and change into inappropriate cell types, tumor formation, blindness, kidney failure, embolic phenomena, and infections because of the contamination of the finished products [55]. Also, most of these establishments offer therapies where the extraction, isolation, and application processes do not comply with standardized protocols (good manufacturing practices of GMPs), which leads to total failure.

To find a way around this problem, regulatory agencies around the world should put pressure on their governments to take more rigorous actions against these dishonest and unprincipled establishments that take advantage of the misfortune of patients and damage the clinical research carried out in this field. For this reason, the population, in general, has been exhorted to do their part in this great problem, first, by making sure that before receiving any therapy that includes MSCs, they verify that it has been approved by the corresponding regulatory agency. If not, they should at least check that the therapy is being tested in a registered clinical trial and ensure that the treatment is being regulated during its production and application. Finally, the population are encouraged to report illegal establishments where unapproved therapies are being used [56].

In conclusion, there is a long way to go regarding treatment with MSCs and unlocking all of its beneficial potential, as any emerging treatment requires multiple efforts from the areas of interest, including researchers at the basic and clinical level, marketing companies, regulatory agencies, and even governments and international agencies, to solve the problems that arise throughout its development. Large parts of the scientific community and the general public are very enthusiastic about the results that these therapies have shown and the scope that they could achieve in the future in the treatment of a wide variety of diseases upon which their effect is being studied.

## Figures and Tables

**Figure 1 pharmaceuticals-16-01334-f001:**
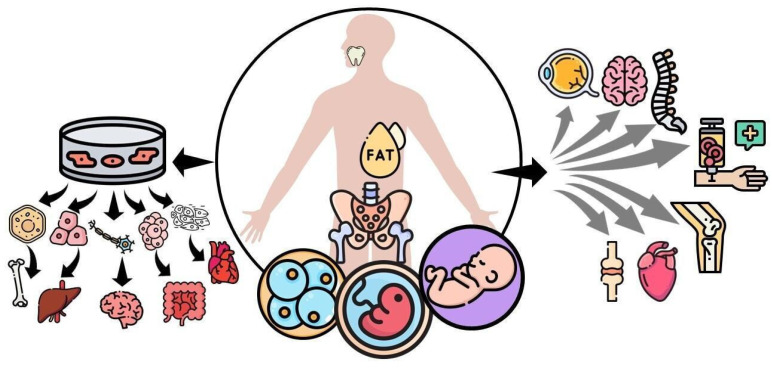
Mesenchymal stem cells (MSCs) can be classified based on the developmental stage in which they are extracted, such as embryonic, fetal, infant, or adult mesenchymal stem cells. They can self-renewal and differentiate into various tissues under appropriate in vitro stimulation, including bone, cartilage, muscle, bone marrow stroma, tendon/ligament, fat, and dermis. On the other hand, MSCs that are not subjected to differentiation in vitro are being tested to treat connective tissue disorders, acute myocardial infarction, amyotrophic lateral sclerosis, spinal cord injury, osteoarthritis, ocular burns, and graft versus host disease.

## Data Availability

Data sharing not applicable.

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
