# Peer review of "Mesenchymal Stem Cell Therapies Approved by Regulatory Agencies around the World"

_pharmaceuticals, 2023, doi:10.3390/ph16091334_

Round 1
Reviewer 1 Report
The review “Mesenchymal Stem Cells Therapies Approved by Regulatory
Agencies Around the World” by Fernández-Garza et al is timely, and I recommend to publish it after a significant revision.
Please find my comments and suggestions below.
Line 44. In the introduction the authors mention some regulatory agencies, and omit others, which may look to the reader as a biased misrepresentation. Obviously, it’s impossible to list all regulatory agencies, but the authors may want to justify their choice. They also may want to refer to the publications describing product classification mentioned in line 44.
Line 129. The authors state that the “performed a systematic review that reported that up until June 2020”. Why do not they preformed a review of trials up until now or until beginning of 2023?
Line 132. There is a typo “with the largest number or registered clinical trials” instead of “with the largest number of registered clinical trials”
It might be of interest to expand a list of conditions/diseases discussed, even with unpulbished results, and include neurology/Spinal Cord Inijury as well (for example, see ClinicalTrials.gov Identifier: NCT04205019, or others), as MSCs are promising therapies for SCI.
Line 176. Please fix the formatting of the text in the first column, to make sure that both words (“Name/year”, etc) fit into one line. Perhaps, changing the size of the text will help.
Lines 236, 241. The authors quote publication [34], however this publication is a review, not the original work. Please refer to the original publications.
Line 248. “Another very similar concern is the so-called medical tourism, which consists of the same situation where people fo to foreign countries” - please paraphrase this sentence and fix the typo
Overall, the review is rather too short and it does not cover the key scientific publications in the field published recently. Indeed, the author quote only 39 sources, of which several sources are not peer-reviewed publications (for example, reference 37. News wise (Articles). Businesses selling non-FDA-approved stem cell products grew four-fold in five years, UCI study
says. [cited 2022 Jan 14]. Available from: https://www.newswise.com/articles/businesses-selling-non-fda-approved-stem-cell-products-grew-four-fold-in-five-years-uci-study-says. ).
I strongly advise the authors to elaborate in writing and add at least a page to discussion.
Author Response
Thank you very much for your observations, we did everything necessary to improve it.
Reviewer 2 Report
This review summarized the current regulatory agencies approved MSCs therapies around the world. The following comments are provided for authors’ consideration:
1. As the title indicated, the focus of this review is “Mesenchymal Stem Cells Therapies Approved by Regulatory Agencies Around the World”. There are several reviews on MSCs therapy had described the MSCs origins and clinical applications, so the part 2 and 3 could be briefer. In part 4, the authors described the 12 agency-approved therapies, which is quite informative. In my view, the readers may expect more detailed pros and cons reported in these 12 therapies. The issues summarized in part 5 is too general, which didn't dig up the detailed shortcomings in these 12 therapies. The discussion on the potential directions is not related to the 12 therapies, and has been reported previously, which makes this manuscript unnecessary. More insightful discussion on these 12 therapies is expected in this review.
2. The authors are suggested to highlight the importance of this review. Why this review is essential compared to previous reviews on MSCs therapies?
3. In Figure 1, the authors may want to exhibit that MSCs could be extracted from different developmental stages and various tissues, and could be differentiated into the desired organs. But the legend is too simple to indicate what the figure illustrates.
4. All content in Part 3 is from reference [7], is it necessary to repeat this part in this review?
5. The word "MSCs-approved treatment" is confusing, what the authors would like to describe is "agency-approved MSCs therapies"?
6. Line 54, what is “Arnold I. Caplan, 53 personal communication 2022”?
Readable English language
Author Response

(The authors gave the same response as above.)

Round 2
Reviewer 1 Report
I appreciate the authors effort to improve the quality of their important manuscript.
However, I still think that the text is far from being flawless.
Please find my comments and suggestions below:
In the beginning of the review the authors state :The totipotent stem cells have the potential to differentiate into any type of cell found in an organism, which is the case of the zygote. The pluripotent stem cells have the potential to give rise to all the cells in the mesoderm, endoderm, and ectoderm layers”. This statement is not complete. The totipotent stem cells can also give rise to (https://www.nature.com/subjects/totipotent-stem-cells)
The authors write “MSCs defining characteristics are the capacity to divide symmetrically or
asymmetrically, motility, cell differentiation, and cell organization”. To me this sentence is meaningless. Indeed, all cells have cell organization, motility etc. It's not a specific defining characteristic. It is impossible to identify MSC based on characteristic such a “motility”, “cell organisation”. These words are just very general words, applicable to almost any living cell. To make myself clear I will use an example. For example, expression of CD8 molecule and CD8-mediated cytotoxicity is a defining characteristic of CD8+ cytotoxic T lymphocyte, it defines and characterises this particular type of cells, while being live cell with cell organisation is not a defining characterystic of CD8+ T lymphocyte.
I also strongly suggest removing the new part of the text, starting from “MSCs defining characteristics are” and ending with “their previous characteristics [17]”. I find it not relevant to the subject and rather incoherent. Instead, the authors may just mention something along the line that there is a heterogeneity of MSCs cell population and MSCs can be a subject of clonal selection due to the environmental changes such as, etc, etc.
The authors state “once an MSC cell enters the bloodstream of the patient being treated, there is no current technology to determine where it docks, what it secretes in the bloodstream or at the docking site, or what cascade of events its presence in the bloodstream or at a particular site will initiate”. - it is a rather vague and arguable statement. The MSC cells can be labeled and tracked (see https://svn.bmj.com/content/6/1/121, https://www.ncbi.nlm.nih.gov/pmc/articles/PMC6955624/), even with the use of FDA-approved labeling molecules (https://www.nature.com/articles/srep26271).
Perhaps the authors of the manuscript might benefit from contacting https://www.mdpi.com/authors/english service.
Author Response
.

Reviewer 2 Report
All my comments have been properly addressed.
Author Response
.